# Insights from the Newborn Screening Program for Very Long-Chain Acyl-CoA Dehydrogenase (VLCAD) Deficiency in Kuwait [note 1]

**DOI:** 10.3390/ijns11010019

**Published:** 2025-02-28

**Authors:** Hind Alsharhan, Amir A. Ahmed, Marwa Abdullah, Moudhi Almaie, Makia J. Marafie, Ibrahim Sulaiman, Reem M. Elshafie, Ahmad Alahmad, Asma Alshammari, Parakkal Xavier Cyril, Usama M. Elkazzaz, Samia M. Ibrahim, Mohamed Elghitany, Ayman M. Salloum, Fahmy Yassen, Rasha Alsafi, Laila Bastaki, Buthaina Albash

**Affiliations:** 1Department of Pediatrics, Health Sciences Centre, College of Medicine, Kuwait University, Safat 13110, Kuwait; 2Department of Pediatrics, Farwaniya Hospital, Ministry of Health, Sabah Al-Nasser 92426, Kuwait; 3Kuwait Medical Genetics Center, Ministry of Health, Sulaibikhat 80901, Kuwait; 4Newborn Screening Laboratory, Kuwait Medical Genetics Center, Ministry of Health, Sulaibikhat 80901, Kuwait; 5Newborn Screening Office, Adan Hospital, Ministry of Health, Hadiya 52700, Kuwait; 6Newborn Screening Office, Farwaniya Hospital, Ministry of Health, Sabah Al-Nasser 92426, Kuwait; 7Newborn Screening Office, Al-Sabah Maternity Hospital, Ministry of Health, Sulaibikhat 80901, Kuwait; 8Newborn Screening Office, Jahra Hospital, Ministry of Health, Jahra 00020, Kuwait; 9Biochemistry Laboratory, Al-Sabah Hospital, Ministry of Health, Shuwaikh 70051, Kuwait; 10Newborn Screening Office, Jaber Al-Ahmad Hospital, Ministry of Health, Kuwait City 72853, Kuwait; 11Department of Pediatrics, Adan Hospital, Ministry of Health, Hadiya 52700, Kuwait

**Keywords:** very long-chain acyl-CoA dehydrogenase deficiency, acylcarnitine, molecular testing, newborn screening, incidence

## Abstract

Newborn screening for very long-chain acyl-CoA dehydrogenase (VLCAD) deficiency in Kuwait was initiated in October 2014. Over a 7-year period (January 2015 to December 2021), 43 newborns were diagnosed with VLCAD deficiency out of 356,819 screened, corresponding to an incidence of 1:8290 and 1:5405 among only Kuwaiti newborns. This study represents the first comprehensive review of newborn screening for VLCAD deficiency in Kuwait. The screening process begins with the detection of elevated blood C14:1 levels in dried blood spots, followed by confirmatory testing using dried blood spots acylcarnitine profiling, with or without molecular testing. Furthermore, this study demonstrates that incorporating the C14:1/C2 ratio as a supplementary marker in first-tier testing alongside C14:1 improves the positive predictive value (PPV) of the current newborn screening for VLCAD deficiency. Adding molecular genetic testing for known VLCAD variants as a second-tier strategy to the national program is also recommended to further enhance specificity and improve PPV. Our findings provide evidence that the expanded newborn screening program in Kuwait has successfully facilitated the early detection of VLCAD deficiency, preventing death and disability in affected infants.

## 1. Introduction

Fat serves as the primary energy reserve and a crucial fuel source for the heart and skeletal muscles [1]. Disorders in fatty acid oxidation (FAO) are metabolic conditions that hinder the body’s ability to utilize fat as an energy source, particularly during fasting. These disorders account for a considerable number of cases of hypoketotic hypoglycemia in early infancy, as well as cardiomyopathies, arrhythmias leading to sudden infant death, and exercise-induced rhabdomyolysis in older children [1,2]. Various metabolic pathways involved in FAO can be affected, classifying them as inborn errors of metabolism (IEM). These include the esterification of carnitine with fatty acyl-CoA ester, the transport of acylcarnitines across the mitochondrial membrane, and the subsequent release of carnitine and fatty acyl-CoA to the several steps of beta oxidation of fatty acids. Newborn screening for FAO disorders has significantly reduced mortality by enabling early intervention, primarily through fasting avoidance [2,3].

Very long-chain acyl-CoA dehydrogenase (VLCAD) is one of the enzymes involved in the mitochondrial ß-oxidation of long-chain fatty acids. It is encoded by the *ACADVL* gene. Pathogenic variants in *ACADVL* result in very long-chain acyl-CoA dehydrogenase deficiency (VLCADD), an autosomal recessive disorder with a worldwide incidence of 1:30,000 to 1:100,000, leading to the accumulation of high blood levels of long-chain acylcarnitine conjugates, especially the tetradecenoyl (C14:1) acylcarnitine [2,4]. VLCADD is a heterogeneous disorder with three main phenotypes, ranging from severe early-onset cardiac and multiorgan failure, hepatic dysfunction with hypoglycemia, to later-onset myopathy with rhabdomyolysis [5]. Early diagnosis results in improved prognosis and has thus been included in many newborn screening (NBS) programs.

The treatment of VLCADD is divided into acute and chronic management strategies. Acute management mainly involves the provision of a sufficient amount of glucose-containing fluids to correct hypoglycemia and inhibit lipolysis, while chronic management mainly relies on avoiding fasting, restricting long-chain fatty acids, and supplementing with medium-chain triglycerides or triheptanoin and carnitine [3,5,6].

Among the various inborn errors of FAO, VLCADD is the most common FAO defect in the Kuwaiti population, which has been particularly noted following the launch of the expanded newborn screening (NBS) program in Kuwait. As previously explained by our group, the publicly funded expanded NBS program in Kuwait was initiated in October 2014 to include a total of 22 endocrine and metabolic disorders via testing dried blood spots (DBS), including VLCADD [7]. Prior to screening for VLCADD, many affected siblings experienced unexplained neonatal and early infantile deaths (based on personal and local experience). However, following the diagnosis of VLCADD via NBS in their newly born affected siblings, it has become clear that VLCADD was the cause of these deaths. There is evidence supporting the benefit of NBS for FAO defects [8]. Our expanded national NBS program has facilitated the early diagnosis of such individuals and subsequently prevented significant morbidity and mortality. This is the first study to review the Kuwait NBS program’s experience in screening for VLCADD since its launch in October 2014. This paper is a revised and expanded version of a paper entitled “Newborn screening experience for very long chain Acyl-CoA Dehydrogenase (VLCAD) deficiency in Kuwait”, which was presented at ACMG Conference online in March 2022 [9].

## 2. Materials and Methods

### 2.1. NBS Registry

A retrospective analysis of the data registry for the NBS was performed over a 7-year period between January 2015 and December 2021 in Kuwait, after obtaining consent from the NBS program at the Kuwait Medical Genetics Center (KMGC) and the Ministry of Health in Kuwait. These data included newborns delivered at both private and public hospitals in Kuwait, according to the percentage of coverage shown in Table 1. Data on metabolite concentrations in dried blood spots (DBS) at the time of screening, obtained from all newborns, were reviewed, and only DBS detecting elevated C14:1 were included in this study.

### 2.2. Initial DBS Collection Protocol

The first DBS samples were collected on Whatman 903 filter papers within 24–72 h of life but could be accepted up to one month after birth. For any DBS collected before 24 h of age, a new DBS would be repeated within 7 days. All initial DBS samples were sent to the NBS laboratory at the KMGC for analysis. Samples showing high levels of C14:1 were followed by a second confirmatory DBS for acylcarnitine analysis, in addition to urine organic acids, which were sent to the Sabah Hospital Biochemistry Laboratory, the main laboratory that provides diagnostic testing for metabolic disorders (Figure 1). Molecular and enzymatic testing are not routinely performed as follow-up.

### 2.3. Analytical Methods

#### 2.3.1. Acylcarnitine Measurement in First (Initial) DBS

The method used to measure C14:1 in the first DBS is a semi-quantitative determination via tandem mass spectrometry (MS/MS) without a butylation step [10,11]. Electrospray ionization tandem mass spectrometry (ESI-MS/MS) analysis was performed using a Waters Triple Quadrupole Mass Spectrometer (Xevo TQD from Waters manufacture). The analytical measurements were performed in multiple reaction monitoring mode (MRM) by using MassLynx and NeoLynx 4.1 software. The stable isotope amino acids and acylcarnitines internal standards, supplied by Chromsystems, measure 11 amino acids, free carnitine, and 30 different acylcarnitines. To monitor the performance of our assays, quality control (QC) was conducted using the same sample plate.

#### 2.3.2. Acylcarnitine Measurement in Second DBS (Confirmatory DBS)

An API 3200 triple quadrupole tandem mass spectrometer (AB-SCIEX) and liquid chromatography–tandem mass spectrometry (LC–MS/MS) were used to analyze acylcarnitine with the butylation step, as previously described with minor modifications [12].

### 2.4. Molecular Testing

A total of 55 newborns with positive screening results for VLCADD (including 38 biochemically confirmed VLCADD cases and 12 with an initial positive screen) underwent clinical genetic testing. This testing was performed using either targeted variant testing via polymerase chain reaction (PCR) amplification followed by Sanger sequencing or next-generation sequencing technology using the Ion AmpliSeq Inborn Errors of Metabolism community panel (Thermo Fisher Scientific, Waltham, MA, USA).

## 3. Results

### 3.1. NBS Registry

The Kuwait NBS data registry included 395,979 samples for 356,819 screened neonates (204,557 = 57% Kuwaiti; 152,262 = 43% non-Kuwaiti) born in Kuwait between January 2015 and December 2021 (Table 1). The number of screened samples is typically higher than the number of screened newborns since the national NBS protocol of Kuwait recommends collecting three DBS for premature babies during the first month of life (at 72 h, 2 weeks, and 1 month of age). Furthermore, DBS that are collected before 24 h are required to be repeated within the first week of life. The age of the newborn at the time of the first DBS collection ranged from 1 to 8 days of age, with a median age of 1.8 days (Table 2).

### 3.2. DBS with Elevated C14:1

About 178 screened newborns had elevated C14:1 with a cutoff value of 0.29 μmol/L on the first DBS (corresponding to the mean + 5SD) for the period between 2015 and 2021. A total of 43 confirmed cases of VLCADD were identified based on elevated blood C14:1 (cutoff 0.29 μmol/L) and C14:1/C2 (cutoff value 0.03), as well as other long-chain acylcarnitines (including C14, C14:2, and C14:1/C16), on their repeated DBS with or without molecular testing from 2015 to December 2021 through the NBS program. Five newborns died before a second confirmatory DBS or genetic testing could be obtained; two of them had siblings born later who were confirmed to have VLCADD. The precise cause of their death remains uncertain, but it may be linked to VLCADD, considering the severity of the phenotype observed in our region, particularly the neonatal cardiomyopathy seen in some affected newborns. The 43 cases (23 male/20 female) were from 39 different families; 38 individuals were Kuwaitis, 2 were Saudi Arabian, and 3 babies were Indian (Table 2). This has resulted in an overall incidence of 1:8290 and 1:5405 among only Kuwaiti newborns. The final NBS result was available at a median age of 4.4 days (range 2–9 days), allowing for the prompt initiation of specific treatment plans at the same median age of 4.4 days. However, an exception was Case 7, where the result was not successfully communicated to the caregivers, leading to a delay in intervention (Table 2).

### 3.3. Evaluation of the C14:1/C2 Ratio as a Potential Strategy for Screening for VLCADD in Kuwait

Using a C14:1/C2 cutoff of 0.03, in addition to the established C14:1 cutoff of 0.29 μmol/L, resulted in 113 suspected VLCADD cases (0.38% of samples) and a specificity of 99.97 (Table 3). This adjustment improved the positive predictive value (PPV) of VLCADD screening from 27 to 42% while maintaining 100% sensitivity. However, applying higher C14:1/C2 ratio cutoffs (e.g., >0.05, >0.08, >0.1) led to further increases in specificity, up to 99.98%, and improved PPV, reaching 65% at a cutoff of >0.1. Notably, these higher cutoffs resulted in the misclassification of three affected newborns as false negatives, reducing sensitivity to 93% (Table 3).

### 3.4. Molecular Genetic Testing Results

Molecular genetic testing was conducted on 38 of the 43 biochemically confirmed VLCADD cases. All but one were found to carry the previously reported pathogenic homozygous nonsense founder variant in exon 2 of the *ACADVL* gene (NM_000018.3): c.65C>A; p.(Ser22Ter) (Table 2). The remaining newborn of Indian ethnicity had a different previously reported pathogenic homozygous variant, c.1269+1G>A (Table 2). Several newborns who initially tested positive for VLCADD were identified as either heterozygous for the founder variant in *ACADVL* (c.65C>A or c.1615G>A; p.(Ala539Thr)) or as compound heterozygous for pathogenic variants in the *ACADVL* gene, along with uncertain significance intronic nucleotide variations in *ACADVL*. These included the pathogenic variants c.1615G>A (p.Ala539Thr), c.65C>A (p.(Ser22Ter)), or c.896_898delAGA (p.Lys299del), along with uncertain significance intronic nucleotide variations in *ACADVL* (NM_000018.3): c.478-116_478-100delCTGCCCTAGGTCAGGAA, c.478-89T>C, and c.1605+6T>C (Table 4 and Table 5).

## 4. Discussion

This study provides the first comprehensive review of the Newborn Screening (NBS) Program for VLCADD in Kuwait since its launch in October 2014. The marker used for VLCADD detection in NBS is elevated C14:1, especially when levels exceed 1 µmol/L [8,13]. Additional markers, including C14:2 and C14:0, as well as the abnormal acylcarnitine ratio (C14:1/C2) measured via LC-MS/MS in DBS, further support the diagnosis, but they are not required for the diagnostic protocol for a positive screen for VLCADD. Follow-up confirmatory testing typically includes plasma acylcarnitine analysis and, when available, molecular and/or enzymatic testing. In Kuwait’s NBS, molecular testing is not routinely included as a standard step in the protocol for positive NBS cases. If the confirmatory acylcarnitine profile (C14:1 and C14:1/C2 ratio) is normal, the case could be dismissed without further molecular genetic testing. As a result, 135 cases were excluded and considered false positives for VLCADD based on normal confirmatory acylcarnitine analysis. However, it is important to note that the normalization of follow-up acylcarnitine profiles does not exclude VLCADD, as the condition may be masked depending on the timing and anabolic state of the newborn during testing [14,15,16,17]. Moreover, the degree of C14:1 elevation in the initial NBS does not necessarily correlate with disease severity or the degree of enzyme deficiency [16]. In Kuwait, VLCADD is suspected when the C14:1 level in the first DBS exceeds 0.29 µmol/L, yielding a positive predictive value (PPV) of 22%.

The addition of the C14:1/C2 ratio as a supplementary marker in newborn screening has the potential to reduce false positive rates, thereby enhancing the overall efficiency of the screening process. Merritt et al. explored additional ratios, such as C14:1/C16 and C14:1/C12:1, finding them useful in differentiating true VLCADD cases from other metabolic disorders [18]. Another study emphasized that the C14:1/C2 ratio is a more sensitive marker than C14:1 alone, aiding in the detection of VLCADD cases [19]. Incorporating a C14:1/C2 cutoff of 0.03 alongside the established C14:1 cutoff of 0.29 µmol/L improved the positive predictive value (PPV) in our study from 27% to 42% while maintaining 100% sensitivity. Higher cutoffs for the C14:1/C2 ratio (e.g., >0.05, >0.08, >0.1) further increased specificity to 99.98% and raised the PPV to 65%, but resulted in reduced sensitivity (93%) due to the misclassification of three affected newborns as false negatives (Table 3).

To further enhance specificity and improve PPV, molecular genetic testing for known VLCADD variants is strongly recommended as a second-tier diagnostic strategy in cases with elevated C14:1 and C14:1/C2 ratios.

The confirmatory DBS cutoff for C14:1, established at 1 µmol/L, applied in our cohort, aligns with the 25th percentile of the disorder range, providing a robust reference for confirming diagnoses [20]. C14:1 levels greater than 0.8 µmol/L suggest VLCAD deficiency but can also be observed in carriers and some healthy individuals without *ACADVL* pathogenic variants [5]. In our cohort, seven heterozygous newborns for the *ACADVL* pathogenic variant p.(Ser22Ter) showed elevated C14:1 levels, ranging from the cutoff of 0.3 to 1.61 nmol/mL, along with an increased C14:1/C2 ratio in four out of seven cases, leading to a positive VLCADD screen result (Table 4). Several newborns who screened positive for VLCADD (five newborns) were found to have compound heterozygous variants in the *ACADVL* gene: a pathogenic c.1615G>A (p.Ala539Thr), c.65C>A p.(Ser22Ter), or c.896_898delAGA (p.Lys299del), along with intronic nucleotide variations in *ACADVL* (NM_000018.3): c.478-116_478-100delCTGCCCTAGGTCAGGAA, c.478-89T>C, and c.1605+6T>C (Table 5). These infants exhibited initial abnormal C14:1 levels with or without normal C14:1/C2 ratios, and their confirmatory acylcarnitine analysis was normal (Table 5). However, all of them remained asymptomatic and continue to be symptom-free while on a regular diet. We identified an additional newborn with an initially abnormal NBS result for VLCADD, but with normal confirmatory acylcarnitine analysis. This individual was later found to be homozygous for these specific intronic nucleotide variations and has remained asymptomatic on a regular diet. However, long-term follow-up for individuals who are either compound heterozygous or homozygous for these intronic variants is needed to further understand their clinical significance. Unfortunately, enzymatic testing could not be performed on these individuals, leaving some diagnostic uncertainty. These intronic nucleotide variations have been previously reported in conjunction with a heterozygous pathogenic missense variant in the *ACADVL* gene, which is associated with adult-onset exercise-induced rhabdomyolysis [21]. These intronic variants were reported as benign in Clinvar (rs359998690) in 2019, although no functional evidence was provided (Accession: VCV001248482.2) [22]. Furthermore, these intronic variants have a high population frequency (GnomAD genomes allele frequency of 0.475) [23].

Several challenges complicate VLCADD screening and diagnosis. Acylcarnitine abnormalities may resolve under physiological wellness conditions, such as after feeding or glucose infusion, leading to the potential misclassification of mild phenotypes [5]. One of the other factors that might cause false positive elevations of C14:1 and C14:1/C2 is severe body weight loss on the sampling day of NBS [5]; however, this was not observed in our cohort.

Although a C14:1 cutoff level greater than 1 µmol/L on an initial NBS test strongly suggests VLCADD [24], affected individuals with levels as low as 0.39 µmol/L were genetically confirmed in this cohort.

An additional challenge faced by any NBS program is the implementation of an effective follow-up protocol, as highlighted by Case 7 (Table 2). This newborn, delivered in a private hospital, was not appropriately followed after a positive NBS result, resulting in a missed diagnosis. The child presented with dilated cardiomyopathy at 3 months of age, requiring admission to the pediatric intensive care unit (PICU) (Table 2). This case underscores the critical importance of establishing robust and standardized follow-up procedures that extend across all healthcare sectors, including private institutions, to ensure timely diagnosis and intervention.

The incidence of VLCADD in Saudi Arabia is about 1:37,000, with the c.848T>C; p.(Val283Ala) pathogenic variant commonly reported but absent in this Kuwaiti cohort [6,7,8,18,25]. Instead, a founder loss-of-function variant in the *ACADVL* gene, c.65C>A; p.(Ser22Ter), was identified in 97% of affected newborns and accounted for 80% of VLCADD-related variants in the Saudi population. This variant is linked to severe clinical presentations and high mortality [4,7,8,10,25,26]. In this cohort, at least eight newborns presented with symptoms prior to receiving NBS results. Additionally, one Indian newborn with a homozygous c.1269+1G>A pathogenic variant exhibited a severe phenotype, and their sibling had succumbed on the first day of life (Table 2).

VLCADD was confirmed in 88% of our positively screened newborns through the identification of biallelic pathogenic variants in the *ACADVL* gene. The incidence of VLCADD in Kuwait in this study is 1 in 8290 and 1:5405 among only Kuwaiti newborns. Since *ACADVL* gene sequencing has not been performed for all babies with abnormal NBS results for VLCADD, the true incidence of VLCADD may be underestimated. Additionally, some affected babies may have passed away shortly after birth, prior to the collection of confirmatory acylcarnitine analysis, and therefore were not included in the NBS registry for confirmed VLCADD cases, potentially contributing to the relatively low incidence. Furthermore, before 2019, some private hospitals sent NBS samples to external laboratories rather than through the Kuwait National NBS Laboratory, which may have affected the accuracy of the reported incidence of VLCADD and other screened disorders.

Consanguinity was observed in all affected individuals in our cohort, with a founder mutation in the *ACADVL* gene, p.(Ser22Ter), identified in all affected Kuwaiti and Saudi individuals. Although all affected Kuwaiti and Saudi newborns shared the same genotype, their clinical outcomes varied; some passed away within the first few days of life, while others survived. This discrepancy may be attributed to differences in the level of care and awareness of VLCADD. The majority of these individuals were born and managed in private hospitals, where experience in handling IEM is limited. In contrast, governmental hospitals in Kuwait have greater expertise in handling a wide range of IEM cases, potentially contributing to improved survival rates. Given the relatively high incidence of VLCADD based on our NBS data and its associated morbidity and mortality risks, it is essential to consider implementing a national premarital carrier screening program to prevent the disease. Additionally, creating a national registry for rare disorders, including VLCADD and other IEM, along with the identified variants such as founder mutations, is a critical resource for developing a long-term health strategy in the country.

## 5. Conclusions

In summary, our study is the first to review the experience of the NBS program for VLCADD in Kuwait. We have demonstrated that our national NBS for VLCADD is highly effective, with positive cases successfully followed and affected infants receiving treatment within an average of 9 days of life. The incidence of VLCADD in Kuwait since the expansion of the NBS program in October 2014 is 1 in 8290, underscoring the program’s effectiveness and significance. Furthermore, we recommend incorporating the C14:1/C2 ratio as a supplementary marker together with C14:1 and adding molecular genetic testing as a second-tier strategy to the national VLCADD program to further enhance the specificity of NBS testing.

## Figures and Tables

**Figure 1 IJNS-11-00019-f001:**
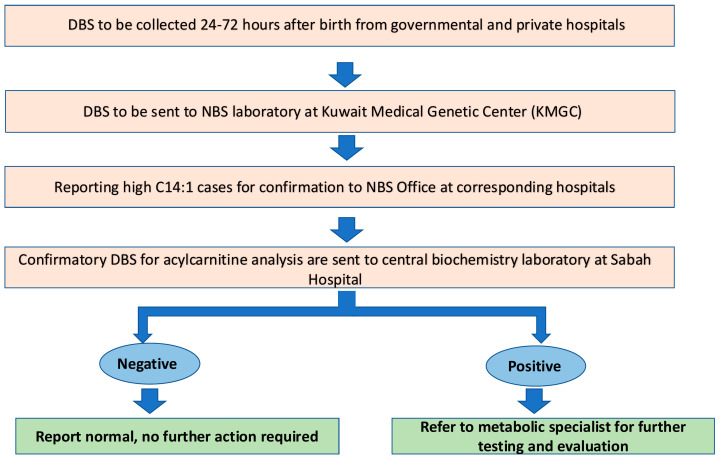
Process of newborn screening for very long chain acyl-CoA dehydrogenase (VLCAD) deficiency in Kuwait. DBS: dried blood spot; NBS: newborn screening.

**Table 1 IJNS-11-00019-t001:** Overview of the numbers of children born in Kuwait and screened through the national Newborn Screening Program over a 7-year period from January 2015 to December 2021.

	2021	2020	2019	2018	2017	2016	2015	Total
Total sample received in the NBS laboratory	57,600	56,441	56,333	55,210	59,655	57,951	52,789	395,979
Total newborns screened	51,585	52,463	50,916	48,501	53,689	52,155	47,510	356,819
No. of all newborns in Kuwait per CSB	51,585	52,463	53,565	56,121	59,172	58,797	59,271	390,974
No. of Kuwaiti newborns per CSB	34,610	31,766	32,263	33,168	33,680	33,431	33,581	232,499
No. of Non-Kuwaiti newborns per CSB	16,975	20,697	21,302	22,953	25,492	25,366	25,690	158,475
Screened Kuwaiti newborns	32,657	30,444	30,145	28,645	29,074	28,733	24,859	204,557
Screened non-Kuwaiti newborns	18,928	22,019	20,771	19,856	24,615	23,422	22,651	152,262
Newborns not screened under the national NBS program	0	0	2649	7620	5483	6642	11,761	34,155
Percent of coverage of national NBS program	100%	100%	95.1%	86.4%	90.7%	88.7%	80.1%	-

CSB: central statistical bureau; NBS: newborn screening; No.: number.

**Table 2 IJNS-11-00019-t002:** Overview of individuals diagnosed with VLCADD via Newborn Screening Program in the period from 2015 to 2021 in Kuwait, demonstrating the time between birth and start of treatment, the pathogenic variants detected in *ACADVL* gene, and the biochemical results.

	Ethnicity	Date of Birth (Month.Year)	Gender	First DBS		Confirmatory DBS C14:1 Cutoff ≥ 1 μmol/L	Age at DBS Collection (Days)	Age at NBS Result (Days)	Age at Start of Treatment (Days)	^¥^ DNA Variant *ACADVL*	Protein Variant	CK (RR < 250 IU/L)	Outcome
				C14:1 (Cutoff 0.29 μmol/L)	C14:1/C2 (Cutoff 0.03)								
1	K	04.2015	F	2.48	0.19	3.44	1	8	8	c.65C>A (Homo)	p.Ser22Ter (Homo)	NA	Uneventful
2	IND	07.2015	F	3.57	NA	NP	1	-	-	-	-	-	Died at age 1 day
3	K	10.2015	F	2.6	0.42	2.587	1	6	6	c.65C>A (Homo)	p.Ser22Ter (Homo)	NA	Uneventful
4	K	04.2016	M	1.8	0.10	4.649	1	5	5	c.65C>A (Homo)	p.Ser22Ter (Homo)	NA	Uneventful
5	K	06.2016	M	2.9	0.34	3.899	<1	3	3	c.65C>A (Homo)	p.Ser22Ter (Homo)	9031	Uneventful
6	KSA	06.2016	F	5.65	0.31	4.6	3	7	7	c.65C>A (Homo)	p.Ser22Ter (Homo)	1267	Uneventful
7	K	07.2016	F	3.66	0.33	NP	NA	missed	90	c.65C>A (Homo)	p.Ser22Ter (Homo)	NA	Loss of follow-up
8	K *	09.2016	M	2.48	0.38	NP	1	4	-	-	-	-	Died at age 3 days
9	K	10.2016	F	3.56	0.29	4.55	1	3	3	c.65C>A (Homo)	p.Ser22Ter (Homo)	NA	Uneventful
10	K	12.2016	M	3.35	0.21	3.97	1	3	3	c.65C>A (Homo)	p.Ser22Ter (Homo)	NA	Uneventful
11	K	03.2017	M	2.7	0.28	4.24	1	3	3	c.65C>A (Homo)	p.Ser22Ter (Homo)	NA	Uneventful
12	K ^#^	09.2017	F	1.6	0.2	0.25 (N) ^#^	1	4	4	c.65C>A (Homo)	p.Ser22Ter (Homo)	NA	Uneventful
13	K	11.2017	M	0.39	0.04	1.63	3	5	5	c.65C>A (Homo)	p.Ser22Ter (Homo)	260	Uneventful
14	K	12.2017	M	1.56	0.12	4.86	1	3	3	c.65C>A (Homo)	p.Ser22Ter (Homo)	5662	Uneventful
15	K	12.2017	M	2.6	0.23	3.88	2	4	4	c.65C>A (Homo)	p.Ser22Ter (Homo)	NA	Uneventful
16	K **	12.2017	F	2.1	0.22	5.884	1	4	4	c.65C>A (Homo)	p.Ser22Ter (Homo)	1506	Uneventful
17	K *	02.2018	M	0.8	0.15	1.6	1	2	2	c.65C>A (Homo)	p.Ser22Ter (Homo)	3000	Uneventful
18	IND	02.2018	F	1.6	0.14	NP	2	5	-	-	-	-	Died at age 2 days
19	IND ^‡^	04.2018	M	1.7	0.18	0.7 (N) ^‡^	<1	4	0	c.1269+1G>A (Homo)	-	NA	Mild ventricular hypertrophy, EF 50%
20	K	04.2018	M	4.2	0.59	4.85	3	6	6	c.65C>A (Homo)	p.Ser22Ter (Homo)	NA	Uneventful
21	K	04.2018	M	1.5	0.34	2.3	2	4	4	c.65C>A (Homo)	p.Ser22Ter (Homo)	NA	Uneventful
22	K	07.2018	M	2.5	0.66	4.319	8	9	9	c.65C>A (Homo)	p.Ser22Ter (Homo)	NA	Uneventful
23	K	08.2018	F	1.9	0.86	6.1	1	4	4	c.65C>A (Homo)	p.Ser22Ter (Homo)	1521	Uneventful
24	K	10.2018	F	1.44	0.18	2.9	1	3	3	c.65C>A (Homo)	p.Ser22Ter (Homo)	NA	Uneventful
25	K	10.2018	M	1.9	0.16	3.1	1	2	2	c.65C>A (Homo)	p.Ser22Ter (Homo)	NA	Uneventful
26	K ^¢^	10.2018	F	2.5	0.23	0.419 (N) ^¢^	1	4	4	c.65C>A (Homo)	p.Ser22Ter (Homo)	NA	Uneventful
27	KSA	01.2019	M	2.66	0.19	NP	2	5	-	-	-	NA	Died at age 4 days
28	K ^	04.2019	F	2.2	0.13	5.49	2	6	6	c.65C>A (Homo)	p.Ser22Ter (Homo)	3000	Uneventful
29	K	08.2019	M	0.97	0.15	NP	2	4	-	-	-	NA	Died at age 3 days
30	K ^Œ^	09.2019	F	1.3	0.17	0.46 (N) ^Œ^	1	3	3	c.65C>A (Homo)	p.Ser22Ter (Homo)	NA	Uneventful
31	K	09.2019	M	4.12	0.24	4.92	3	6	6	c.65C>A (Homo)	p.Ser22Ter (Homo)	NA	Uneventful
32	K	10.2019	F	3.73	0.22	5.96	3	5	5	c.65C>A (Homo)	p.Ser22Ter (Homo)	1813	Uneventful
33	K	06.2020	M	2.01	0.33	4.69	1	4	4	c.65C>A (Homo)	p.Ser22Ter (Homo)	1418	Uneventful
34	K	06.2020	M	2.01	0.50	2.7	5	7	7	c.65C>A (Homo)	p.Ser22Ter (Homo)	591	Uneventful
35	K	07.2020	M	1.99	0.24	5.25	3	5	5	c.65C>A (Homo)	p.Ser22Ter (Homo)	NA	Uneventful
36	K	08.2020	F	2.03	0.31	1.07	1	3	3	c.65C>A (Homo)	p.Ser22Ter (Homo)	NA	Uneventful
37	K **	09.2020	F	2.17	0.26	3.99	2	3	3	c.65C>A (Homo)	p.Ser22Ter (Homo)	1440	Uneventful
38	K	10.2020	F	0.83	0.10	NA	2	5	5	c.65C>A (Homo)	p.Ser22Ter (Homo)	NA	Uneventful
39	K ^§^	11.2020	F	1.57	0.19	0.54 (N) ^§^	1	3	3	c.65C>A (Homo)	p.Ser22Ter (Homo)	12,000	Uneventful
40	K	03.2021	M	1.21	0.26	3.253	2	6	6	c.65C>A (Homo)	p.Ser22Ter (Homo)	NA	Uneventful
41	K	04.2021	F	2.01	0.49	3.824	1	6	6	c.65C>A (Homo)	p.Ser22Ter (Homo)	NA	Uneventful
42	K ^	06.2021	M	1.78	0.5	3.85	3	4	4	c.65C>A (Homo)	p.Ser22Ter (Homo)	NA	Uneventful
43	K	08.2021	M	1.99	0.21	1.59	2	5	5	c.65C>A (Homo)	p.Ser22Ter (Homo)	1800	Uneventful

CK: creatine kinase; DBS: dried blood spots; EF: ejection fraction; F: female; Homo: homozygous; IND: Indian; K: Kuwaiti; M: male; N: normal; NA: not available; NBS: newborn screening; NP: not performed; RR: reference range; KSA: Kingdom of Saudi Arabia. ^¥^ The reference transcript is (NM_000018.3). ‡ This individual is a sibling to individual (2) and was started on treatment (monogen) at age of 21 h. * Individual (8) and individual (17) are siblings. Individual (17) was started on treatment (monogen) immediately after the abnormal NBS result. # This individual was started on treatment before obtaining confirmatory acylcarnitine profile. ^§^ This individual has an older sibling with VLCADD and was started on treatment immediately after the abnormal NBS result. ** Individual (16) and individual (37) are siblings. ¢ Individual (26) has two older siblings with VLCADD. ^ Individual (28) and individual (41) are siblings. ^Œ^ Individual (30) had an episode of hypoglycemia at age 33 h. He had a normal C14:1 on second confirmatory DBS.

**Table 3 IJNS-11-00019-t003:** Results of applying C14:1/C2 ratio to the current NBS strategy of measuring C14:1 in DBS as first-tier testing, with cutoffs trimmed for 100% sensitivity for all NBS samples in the period from 2015 to 2021.

Cutoff	No. (%) of Positives	False Negative	True Positive	PPV %	Sensitivity (%)	Specificity (%)
C14:1 > 0.29 μmol/L	178	0	43	27	100	99.96
C14:1 > 0.29 μmol/L + C14:1/C2 >0.03	113	0	43	42	100	99.97
C14:1 > 0.29 μmol/L + C14:1/C2 >0.05	85	3	40	60	93	99.98
C14:1 > 0.29 μmol/L + C14:1/C2 >0.08	74	3	40	61	93	99.98
C14:1 > 0.29 μmol/L + C14:1/C2 >0.1	69	3	40	65	93	99.98

No.: number; PPV: positive predictive value.

**Table 4 IJNS-11-00019-t004:** Overview of some of heterozygous individuals for variants in *ACADVL* gene with false positive screening results for VLCADD in the period from 2015 to 2021 in Kuwait.

Ethnicity	Date of Birth (Month.Year)	Gender	DBS		Confirmatory DBS C14:1 Cut-off ≥ 1 mmol/L	Age at NBS Result (Days)	DNA Variant	Protein Variant	CK (RR < 250 IU/L)
			C14:1 (Cutoff 0.29 nmol/mL)	C14:1/C2 (Cutoff 0.03)					
K	11.2018	M	0.39	0.02	normal	3	c.65C>A (Het)	p.(Ser22Ter) (Het)	NA
K	11.2019	F	0.43	0.02	normal	3	c.65C>A (Het)	p.(Ser22Ter) (Het)	358
K	08.2020	M	1.61	0.17	4.1	5	c.65C>A (Het)	p.(Ser22Ter) (Het)	1351
Afghani	09.2020	M	0.35	0.03	normal	4	c.1615G>A (Het)	p.(Ala539Thr) (Het)	144
K	03.2021	M	0.47	0.05	normal	3	c.65C>A (Het)	p.(Ser22Ter) (Het)	1354
K	03.2021	M	0.3	Normal	0.27 (normal)		c.65C>A (Het)	p.(Ser22Ter) (Het)	NA
K	0.6.2021	M	0.36	0.04	normal	3	c.65C>A (Het)	p.(Ser22Ter) (Het)	398

CK: creatine kinase; F: female; Het: heterozygous; K: Kuwaiti; NA: not available; M: male; NBS: newborn screening; RR: reference range.

**Table 5 IJNS-11-00019-t005:** Overview of individuals with positive screening results for VLCADD in the period from 2015 to 2021 in Kuwait found to have compound heterozygous variants in *ACADVL* gene, including a pathogenic c.1615G>A (p.Ala539Thr), c.65C>A p.(Ser22Ter), or c.896_898delAGA(p.Lys299del), along with intronic variant of uncertain significance nucleotide variations in *ACADVL* (NM_000018.3): c.478-116_478-100delCTGCCCTAGGTCAGGAA, c.478-89T>C, and c.1605+6T>C.

Ethnicity	Date of Birth (Month.Year)	Gender	DBS		DBS C14:1 Cutoff ≥ 1 μmol/L	DNA Variant	Protein Variant	CK (RR < 250 IU/L)	Symptoms
			C14:1 (Cutoff 0.29 μmol/L)	C14:1/C2 (Cutoff 0.03)					
Afghani	0.9.2020	M	0.35	0.03	0.22	c.1615G>A (Het)	p.Ala539Thr (Het)	NA	Asymptomatic
K	09.2021	F	0.32	0.04	0.18	c.896_898delAGA	p.Lys299del	NA	Asymptomatic
K	08.2020	F	0.44	normal	0.86	c.65C>A (Het)	p.Ser22Ter (Het)	NA	Asymptomatic
K	08.2020	F	0.44	0.02	0.86	c.65C>A (Het)	p.Ser22Ter (Het)	1351	Asymptomatic
K	0.3.2021	M	0.3	normal	0.271	c.62G>A (Het)	p.Ser21Asn (Het)	NA	Asymptomatic

CK: creatine kinase; F: female; Het: heterozygous; K: Kuwaiti; NA: not available; M: male; RR: reference range.

## Data Availability

The data analyzed during the current study are not publicly available to preserve individuals’ privacy. All data included in this study can be shared upon request to the corresponding author (hind.alsharhan@ku.edu.kw).

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
