# Peer review of "Insights from the Newborn Screening Program for Very Long-Chain Acyl-CoA Dehydrogenase (VLCAD) Deficiency in Kuwait†"

_2409-515X, 2025, doi:10.3390/ijns11010019_

Round 1

Reviewer 1 Report

Comments and Suggestions for Authors

This paper on VLCADD from Kuwait provides valuable insights into the diagnosis and genetic landscape of this rare metabolic disorder in the context of the Kuwaiti population. The findings contribute significantly to the understanding of VLCADD and its clinical presentation, particularly in an area where this disease may not have been thoroughly studied. The study also adds to the global body of research on VLCADD by providing data on screening results, molecular testing, and biochemical markers that could improve diagnostic accuracy and facilitate earlier interventions. Publishing this paper will help broaden the understanding of VLCADD, promote awareness, and support the development of better diagnostic tools and treatment strategies for rare metabolic diseases. Here are my comments

Abstract

The abstract needs to be more concise and shorter. Additionally, it should be reorganized for better clarity. The authors may consider moving the sentence: 'Molecular testing was conducted for 38 of these newborns, revealing a previously reported pathogenic homozygous nonsense variant in exon 2 of the ACADVL gene, c.65C>A; p.(Ser22Ter), in all but one case. The remaining newborn was found to carry a different previously reported pathogenic homozygous variant, c.1269+1G>A.' to follow the phrase '... with or without molecular testing.

Introduction

Paragraph 1, Line 4: The authors may consider adding additional clinical features related to fatty acid oxidation (FAO) disorders and including more citations to provide a comprehensive overview.

Paragraph 1, Line 9: The authors should clarify how FAO disorders lead to death before introducing the statement: 'Newborn screening for such disorders has resulted in the prevention of many deaths...'

Paragraph 1, Line 9: Since there are many types of FAO disorders, the authors should be more specific when discussing treatment. Some FAOD patients may present with rhabdomyolysis or liver failure. The authors could specify by stating, 'Acute management for hypoglycemia,' as hypoglycemia is the most common symptom related to FAOD, among other acute manifestations.

Paragraph 1, Line 12: The discussion of chronic management should specify which FAO disorder is being addressed. Not all FAO disorders require restriction of long-chain fatty acids or supplementation with MCT, triheptanoin, and carnitine. If this statement refers to the treatment of VLCADD, please clarify that these interventions are specific to VLCADD.

Paragraph 3: The authors should consider adding citations to support the statement, 'Prior to screening for VLCADD, many affected siblings had unexplained neonatal and early infantile death.' If this observation is based on personal experience, please specify that as well.

Result

Table 2

Could the authors clarify what is meant by 'missed NBS' in the outcome of patient 7. The authors may use loss of follow-up

3.2. DBS with elevated C14:1

The authors stated that among 178 newborns with C14:1 levels above 0.29, 43 cases were diagnosed with VLCADD based on C14:1 and C14:1/C2 values on confirmatory DBS. Please include the specific cutoff values for C14:1 and C14:1/C2.

Since molecular testing was not performed in all 178 cases, how did the authors rule out VLCADD in135  cases where confirmatory DBS results were normal? It should be noted that the normalization of follow-up acylcarnitine profiles does not necessarily exclude VLCADD

A total of 43 confirmed cases of VLCADD were identified based on elevated blood C14:1 and C14:1/C2 levels 'as well as other long chain acylcarnitines. Since VLCADD patients typically exhibit elevations in specific long-chain fatty acids, the authors should either specify these or consider removing the phrase 'as well as other long-chain acylcarnitines……

Five newborns died. What was the cause of death—was it unknown or related to VLCADD? Please provide additional details.

Discussion

Paragraph 1, line 4: authors should specify what ratio are used to support the diagnosis

Author should also discuss how the author exclude VLCAD in cases 135 cases when confirmatory DBS C14:1 and/or C14:1/C2 are normal.

Paragraph 4: Several falsely positive screened newborns for VLCADD (five newborns) were found to have compound heterozygous variants. Please clarify in the 'Methods' section how many cases of positive NBS for VLCADD underwent molecular testing. It was mentioned in the 'Molecular Testing' section that most of the biochemically confirmed VLCADD cases (38 cases) (Table 2) underwent clinical genetic testing, either through targeted variant testing. However, in Paragraph 4, it is stated that several false-positive cases had molecular testing. The authors should clarify this inconsistency.

Are there other studies that use the C14:1/C2 ratio in diagnosing VLCADD? Please reference them and explain how your results compare with those of other studies that used the C14:1/C2 ratio. Are there other diagnostic markers for this disease? Did the authors analyze those markers? If not, please provide the rationale for not analyzing them.

How is the intronic variant classified—pathogenic or VUS?

If newborns were found to have compound heterozygous pathogenic variants, how did the authors conclude that these cases were false positives? Among these, some had elevated CK

Page 8, Paragraph 4: Since genetic testing has not been obtained...' The authors may consider revising this to 'Since molecular testing or ACADVL sequencing has not been obtained...' to be more specific.

Comments on the Quality of English Language

Make the manuscript clearer, more concise, and eliminate redundancy.

Author Response

Thank you so  much for your valuable feedback and review. Kindly see attached document for the responses for your important comments.

Reviewer 2 Report

Comments and Suggestions for Authors

â—Ž General Comment

In this manuscript entitled “Insights from the Newborn Screening Program for Very Long Chain Acyl-CoA Dehydrogenase (VLCAD) Deficiency in Kuwait”, the authors conducted the first review of the Kuwait NBS program experience in screening for VLCADD, demonstrating its effectiveness while also discussing its challenges and proposing potential solutions.

Several points need however clarification and a more profound discussion:

â—Ž Specific Comment

〇 Major Comment

  • First of all, the DBS collection protocol is not fully comprehensible based solely on the manuscript content. It would be preferable to include a flowchart or other visual aids for clarification. Additionally, there are several questions regarding the DBS collection protocol.

・The definitions of Second DBS and Confirmatory DBS are ambiguous, and a more detailed explanation is necessary.

・Section “2.2. DBS Collection Protocol” states "For any DBS collected before 24 hours of age, a second DBS would be repeated within 7-day," while section “3.1. NBS registry” states "DBS that are collected before 48 hours are required to have a second DBS collected within the first week of life." Please clarify which description is correct.

・In that case, will the second DBS be tested as a confirmatory DBS?

・For full-term infants, if the First DBS is collected after 48 hours of life and no abnormalities are detected in NBS, is it correct to interpret that the Second DBS is not collected ?

・What are the recommended time points for the three DBS collections in preterm infants?

・In that case, are the samples collected at the second and third time points both tested using the method applied for the second test (2.3.2. Acylcarnitine Measurement in Second DBS)?

  • It is recommended to clearly illustrate the current diagnostic process for VLCADD using a flowchart or other visual aids. Additionally, there are several questions regarding the diagnostic process for VLCADD.

・The manuscript concludes by proposing the addition of criterion C14:1/C2 as a new screening standard. However, based on the text, it appears that criterion C14:1/C2 is already being used in the current screening process for diagnosis. Due to the lack of a detailed description of the diagnostic protocol, it could be interpreted that the protocol did not originally include it, but its use was adopted based on previously reported utility. Alternatively, this manuscript could also be perceived as aiming to establish a new cutoff value for C14:1/C2. It is necessary to clearly define the final objective of this manuscript.

・Section “2.2. DBS Collection Protocol” states " Samples showing high levels of C14:1 were followed by a second DBS for acylcarnitine analysis and urine organic acids." Is there a discussion on urinary organic acid analysis?

・The cases in Table 5 have acylcarnitine measured in plasma. Is there a specific reason why only these cases are reported in plasma? Additionally, under the current diagnostic protocol, in what situations is plasma measurement performed? Please provide a clear explanation of this point as well.

〇Minor Comment

Discussion

・Fourth Paragraph Line 3

C14:1 levels greater than 0.8 “mmol/L”

Is this expression a mistake for μmol/L?

・Fifth Paragraph Line 3

The manuscript states that no severe weight loss, which could potentially cause false positives, was not observed at the time of NBS sampling in your cohort. What is the reason for this?

Table 5

It is necessary to specify the combination of variants present in each individual with compound heterozygosity.

Author Response

We appreciate your valuable review and feedback. Kindly see attached word document, addressing your important review comments and edits.

Reviewer 3 Report

Comments and Suggestions for Authors

Introduction: well written. Paragraph 3, sentences "Prior to screening for VLCADD, many affected siblings had unexplained neonatal and early infantile death. However, following the diagnosis of VLCADD via NBS in their newly born affected siblings, it has been clear that VLCADD was the culprit behind their deaths." There is no citation for these cases. Is this just observation in this general population? Other similar situations in the literature to support this?

Methods 2.4 add space between Thermo Fisher 

Table 2 legend: large I for individual 26

Results: well written and good analysis. 

Discussion: why did you choose to look at the C14:1/C2 ratio. Other ratios for VLCAD (C14:1/C12:1, C14:1/C16). Cite paper(s) that show the benefit of the C14:1/C2 ratio compared to just C14:1 alone. 

Patient population is primarily homozygous for the same variant, but the ACP and presentation are variable. Some babies die within the first few days of life, others survive with mild symptoms. Can you explain the difference in these presentations (prematurity, weight, illness, ect.) that may explain the variability in the genotype/phenotype correlation. A short case description for these patients would be helpful or additional column in Table 2.

Overall, very interesting paper and cohort of patients. VLCADD is important to screen for and this further helps to determine the prevalence of the disorder in addition to more effective newborn screening moving forward. 

Author Response

We appreciate your valuable review and feedback. Kindly see the attached word document, addressing your important reviews and comments.

Round 2

Reviewer 3 Report

Comments and Suggestions for Authors

The expanded introduction is much improved and adds excellent context. 

Materials and methods are clearer.

Results are clearer with added context of which categories patients fall into, cutoffs listed in text, and additional information. Additional molecular data in Table 4 and 5 helps to tease out the positive screened population and understanding the false positive rates.

Discussion is much clearer with additional information added in for which acylcarnitine species and ratios are being monitored and why for the NBS program. The additional explanation of consanguinity and hospital structure in outcomes is also helpful and necessary. 

Minor comments

Line 56: Fatty is capitalized and should not be. 

Line 60: Citation formatting issue.